# In silico assessment of human Calprotectin subunits (S100A8/A9) in presence of sodium and calcium ions using Molecular Dynamics simulation approach

**Nematollah Gheibi[1], Mohammad Ghorbani[2], Hanifeh Shariatifar[3], Alireza Farasat[1]\***

**1** Cellular and Molecular Research Center, Qazvin University of Medical Sciences, Qazvin, Iran,
**2** Department of Nanobiotechnology/ Biophysics, Faculty of Biological Science, Tarbiat Modares University, Tehran, Iran, **3** Young Researchers and Elite Club, Tehran Medical Sciences, Islamic Azad University, Tehran, Iran

\* Farasat8@gmail.com

**Data Availability Statement:** All relevant data are within the manuscript and its Supporting Information files.

## Abstract

Calprotectin is a heterodimeric protein complex which consists of two subunits including S100A8 and S100A9. This protein has a major role in different inflammatory disease and various types of cancers. In current study we aimed to evaluate the structural and thermodynamic changes of the subunits and the complex in presence of sodium and calcium ions using molecular dynamics (MD) simulation. Therefore, the residue interaction network (RIN) was visualized in Cytoscape program. In next step, to measure the binding free energy, the potential of mean force (PMF) method was performed. Finally, the molecular mechanics Poisson-Boltzmann surface area (MMPBSA) method was applied as an effective tool to calculate the molecular model affinities. The MD simulation results of the subunits represented their structural changes in presence of $Ca^{2+}$. Moreover, the RIN and Hydrogen bond analysis demonstrated that cluster interactions between Calprotectin subunits in presence of $Ca^{2+}$ were greater in comparison with $Na^+$. Our findings indicated that the binding free energy of the subunits in presence of $Ca^{2+}$ was significantly greater than $Na^+$. The results revealed that $Ca^{2+}$ has the ability to induce structural changes in subunits in comparison with $Na^+$ which lead to create stronger interactions between. Hence, studying the physical characteristics of the human proteins could be considered as a powerful tool in theranostics and drug design purposes.

## Introduction

Calprotectin is a heterodimeric protein complex which consists of two subunits including S100A8 and S100A9. This protein has a great role in different inflammatory disease and various types of cancers. It is a heterodimeric complex of EF-hand $Ca^{2+}$ binding protein. The protein has two subunits which called S100A8 (MRP8/ Calgranulin A) and S100A9 (MRP14/ Calgranulin B) and are fundamentally expressed by myeloid cells. Thus, they are called

**Funding:** The authors received no specific funding for this work.

**Competing interests:** The authors have declared that no competing interests exist.

Myeloid Related Proteins (MRP) [1]. The S100A8 and S100A9 subunits consist of 93 and 113 amino acid residues with the molecular mass of 10.8 and 13.2 kDa respectively [2]. Both of the subunits have two EF-hands (EF-hand I and II) with the structural motifs of helix-loop-helix. The EF-hand I (N terminal motif) comprising helix I, the non-canonical $Ca^{2+}$ binding loop and helix II while, EF-hand II (C terminal) comprises helix III, the canonical binding loop and helix IV. Thus, this structure provides two $Ca^{2+}$ binding sites which are characterized by lower affinity of EF-hand domain I than EF-hand domain II. Despite the high structural similarity, discrepancies in residue compounds of their $Ca^{2+}$ binding loop provide the EF-hand domains various affinities for $Ca^{2+}$. While main chain carbonyl groups play the major role in binding EF-hand I, the EF-hand II binding carries out via acidic side chains with higher affinity for $Ca^{2+}$. The linker regions of various lengths have the ability to separate two EF-hands in both proteins [3, 4]. In all S100 proteins family, the C-terminal tail of the S100A9 is the longest one [5]. Although the homology of the amino acid sequences between S100A8 and S100A9 is very low, but the tertiary structures of the different monomers are significantly similar which proves that all S100 proteins contain highly conserved structures [6]. In presence of $Ca^{2+}$, the formation of heterodimer complex via non-covalent interactions is more probable. The process of $Ca^{2+}$ binding in each subunits provides the structural conformation in each EF-hand helices. Thus, promoting the structural changes can lead to exposure of hydrophobic residues specially both alpha helices; H-I and H-IV in N and C terminal to create non-covalent interactions including hydrophobic bonding [7, 8]. The $(S100A8/A9)_2$ heterotetramer is specifically created at increased level of $Ca^{2+}$ [9]. Furthermore, some changes in different ionic concentration regulate several cellular procedures and also changes in cytosolic calcium concentration lead to various cellular processes modulation too. The $Ca^{2+}$ binding proteins are considered as key molecules in cell cycle monitoring, differentiation and signal transduction. The aforementioned proteins have the ability to interact with target proteins in a calcium dependent manner which are generally a part of signaling pathways which may lead to create particular cellular responses [10]. The Calprotectin has also been mentioned as DAMPs (Damage Associated Molecular Patterns) with the ability of immune mechanism regulation including inflammatory processes, immune cell migration, immune cell adhesion and responses [11]. Previous studies have provided three specific cell surface proteins as functional receptors for S100A8 and S100A9 such as: Receptor for Advanced Glycation End Products (RAGE) [12], Toll-like receptor 4 (TRL4) [13] and Extracellular Matrix Metalloprotease Inducer (EMMPRIN) [14], even though data are conflicting. Moreover, the Calprotectin has been considered as an endogenous ligand when it is secreted by activated neutrophils to extracellular space which can lead to activate various signaling pathways including JAK/stat, PI3 kinase and MAP kinases (ERK, P38 and JNK). The expression of S100A8/A9 seems to be tissue and cell specific which is differentially increased in various abnormal conditions such as: Inflammatory Bowel Disease (IBD), Rheumatoid Arthritis (RA), Cystic Fibrosis (CF), different types of cancers and neurodegenerative disorders [15, 16]. The Molecular Dynamics (MD) simulation process has a major role in recognition of protein-ligand interactions and the protein conformational modifications at the atomic level. In this study, the MD simulation was applied to evaluate the structural and thermodynamic characteristics of S100A8, S100A9 monomers and S100A8/A9 dimer in physiologic concentration of calcium and sodium ions and also to explain the mechanism of S100A8/A9 dimerization. Although the structural changes were considered experimentally, nevertheless some parts of their mechanism of action were not examined thoroughly. In order to investigate thermodynamics and conformational changes of Calprotectin in different ionic conditions, the MD simulation was done. Additionally, the stability of Calprotectin complex was investigated by Molecular Mechanics Poisson-Boltzmann Surface Area (MMPBSA) and Potential of Mean Force (PMF) methods. Considering the importance of Calprotectin in

inflammation process and various disease, a better understanding of the structure and its conformational changes could be useful in treatment purposes [15].

## Methods

### MD simulation

In current study, the MD simulation process was performed using GROMACS program version 5.1 and the CHARMM 36 force field was used for all simulations. The A9 (ID: 5I8N) [17] and A8 (ID: 5HLO) [18] proteins were obtained by RCSB Protein Data Bank (PDB). In this study, the above systems were solvated by transferable intermolecular potential with 3 points (TIP3P) water model in a cubic box with a distance of 10 Å from the furthest atom of the protein. After solvation, $Na^+$ and $Cl^-$ ions were inserted to neutralize the system. Then, the concentration of 150 mM NaCl and CaCl2 were added to the systems [4, 19] and the energy minimization was carried out using the steepest descent method. Each system was equilibrated by 1 ns MD simulation in the canonical (NVT) ensemble and 1 ns MD simulation in the isothermal–isobaric (NPT) ensemble using position restraints on the heavy atoms of the protein to allow for the equilibration of the solvent. The Nose–Hoover thermostat constant was used for fixing the temperature of the system at 310 K. To maintain the pressure of the system at fixed 1 bar pressure, the Parrinello–Rahman pressure coupling method was used [20]. The electrostatic interactions were calculated using the Particle Mesh Ewald (PME) method with 1.0 nm short-range electrostatic and van der Waals cutoffs [21, 22]. To evaluate the effects of calcium ion on dimerization process, the A9 with and without calcium was used. Thus, the orientation of the complex (A8/A9, ID: 1XK4) [9] was applied to locate the above subunits (ID: 5I8N, ID: 5HLO) appropriately besides each other. Consequently, the process of 200 and 100 ns MD simulation was carried out for dimer (A8/A9) and monomers (each subunits) with time steps of 2 fs on equilibrated systems respectively. A summary of the MD simulation runs was shown in S1 Table.

### RIN calculation

To calculate the residue interaction network (RIN), the average structure of last 10 ns of trajectory system was used. The ring web server [23] was applied to determine the bonds and noncovalent interactions including pi-pi stacking and electrostatic interactions. The Cytoscape program [24] was used to represent the RIN in which the protein amino acids and the bonds are exhibited as nodes and edges.

### PMF

Umbrella sampling (US) is a method which could be applied to obtain the free energy pattern which is often referred as PMF along a special reaction coordinate including protein-protein separation distance. Applying physical reaction coordinate can cause further structural insights [25]. In this study the binding energy of A8 and A9 proteins was calculated from PMF via US method. In the first step, the MD simulation was done to drive A9 far away from A8 which was stable during the simulation process. In the next step about 50 configurations were created along the z axis coordinate. The z coordinates of COM (center of mass) interval among A8 and A9 differed by 0.5 Å in each configurations with the force constant of 10 kcal/mol.Å). The equilibration process for each window was done in a period of 10 ns. Moreover, a 10 ns production run was continued for sampling [26]. Consequently, the PMF pattern was obtained by using Weighted Histogram Analysis Method (WHAM) which was performed via GROMACS software as 'gmx_wham' command [27].

## MMPBSA calculation

The MMPBSA is used as an effective tool to study the molecular models affinities including ligand-protein and protein-protein interactions [28, 29]. To evaluate the binding affinity of S100A8 to S100A9, the binding free energies were measured via MMPBSA to complete the structural analysis. The MMPBSA calculation were carried out on the last 10 ns trajectories via g-mmpbsa tool in GROMACS software [30]. The model of non-polar solvation is based on the solvent-accessible surface area (SASA) with the probe radius 1.4 Å [31]. To understand the MD process efficiently, the root mean square deviation (RMSD), root mean square fluctuation (RMSF), SASA and H-bond of each system were analyzed via GROMACS accessible tools during the simulation process. Consequently, all the images were visualized by Pymol software [32].

# Results and discussion

## Analysis the structural properties of A8, A9 and A8/A9 complex

The RMSD is an essential parameter in predicting the system equilibration during the simulation process [33–35]. To investigate the stability of the structure, the RMSD of Cα was calculated. The RMSD of A8 and A9 in presence of $Na^+$ and $Ca^{2+}$ were analyzed during the simulation process. The RMSD of A8 protein in ionic condition was shown in Fig 1A. As depicted in the figure, there was no difference between the RMSD value of A8 protein in presence of $Na^+$ and $Ca^{2+}$. The RMSD value of the A8 protein in both ionic conditions was nearly similar. The A8 protein in both systems was equilibrated after 10 ns. The Fig 1B, illustrates the A9 protein in sodium and calcium ionic conditions respectively. The results demonstrate that the A9 protein was equilibrated in presence of $Ca^{2+}$ and $Na^+$ after 20 ns of MD simulation. The RMSD average for A9 protein in presence of $Ca^{2+}$ and $Na^+$ was almost 4Å and 2Å respectively. The results showed that the RMSD value of A9 in presence of $Ca^{2+}$ was higher than $Na^+$ which confirmed that the A9 protein acquired significant structural modifications in presence of $Ca^{2+}$ than $Na^+$. The RMSD value of A8/A9 complex in presence of $Na^+$ and $Ca^{2+}$ was illustrated in Fig 1C. As shown in the figure, the complex was equilibrated in presence of $Na^+$ after 20 ns and $Ca^{2+}$ after 50 ns. The RMSD average of the complex in presence of $Na^+$ and $Ca^{2+}$ were 12 Å and 3 Å respectively. These findings confirm that the RMSD value of the complex in presence of $Ca^{2+}$ was lower than the time which the complex was in sodium condition. This was due to the strong interaction between two subunits in presence of $Ca^{2+}$ which led to the complex stabilization.

The same as RMSD, the Radius of gyration (Rg) reveals the equilibrated manner of the protein and also the protein contraction during the simulation [21, 36, 37]. Thus, in current study, the Rg results confirm that in presence of $Na^+$, the A9 protein is more compact than the time which the complex is in presence of $Ca^{2+}$ (Fig 2A). These findings verify that broad structural changes are occurred in A9 protein in presence of $Ca^{2+}$. The Rg results in presence of $Na^+$ and $Ca^{2+}$ in A8, were shown in Fig 2B. The Rg results of the aforementioned protein in presence of $Na^+$ and $Ca^{2+}$ were almost similar which revealed that the protein contraction in both conditions was similar. The Fig 2C shows the Rg results in presence of $Na^+$ and $Ca^{2+}$. As illustrated, the complex was more condensed in presence of $Ca^{2+}$ than $Na^+$.

The Root-mean-square fluctuations (RMSF) of Cα atoms can provide direct insights in to the protein's flexibility and structural fluctuations [38, 39]. To study the flexibility of A9 protein in presence of $Na^+$ and $Ca^{2+}$, the RMSF was calculated. The RMSF of A9 protein in calcium and sodium ionic conditions was shown in Fig 3A. As depicted in the figure, the A9 protein has been reached the higher flexibility in 25–30 and 68–71 amino acid regions in

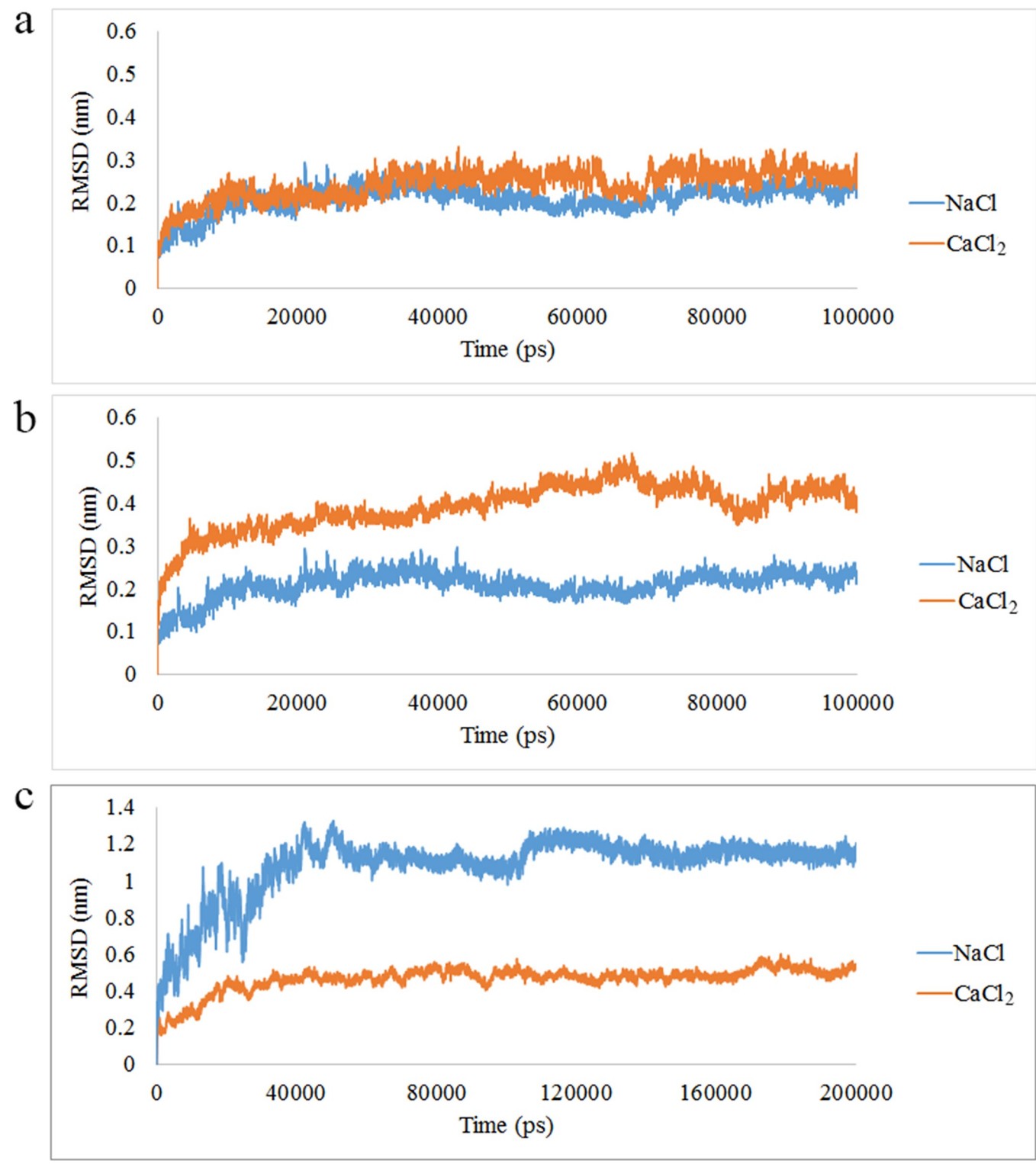

**Fig 1. The root mean square deviation (RMSD) value in presence of Na$^+$ and Ca$^{2+}$.** (a) A8 (b) A9 (c) A8/A9 complex.

presence of Ca$^{2+}$ than Na$^+$. Based on these results, in presence of Ca$^{2+}$, the RMSF value of A9 has a little increase in above regions which are probably the calcium binding sites. It seems

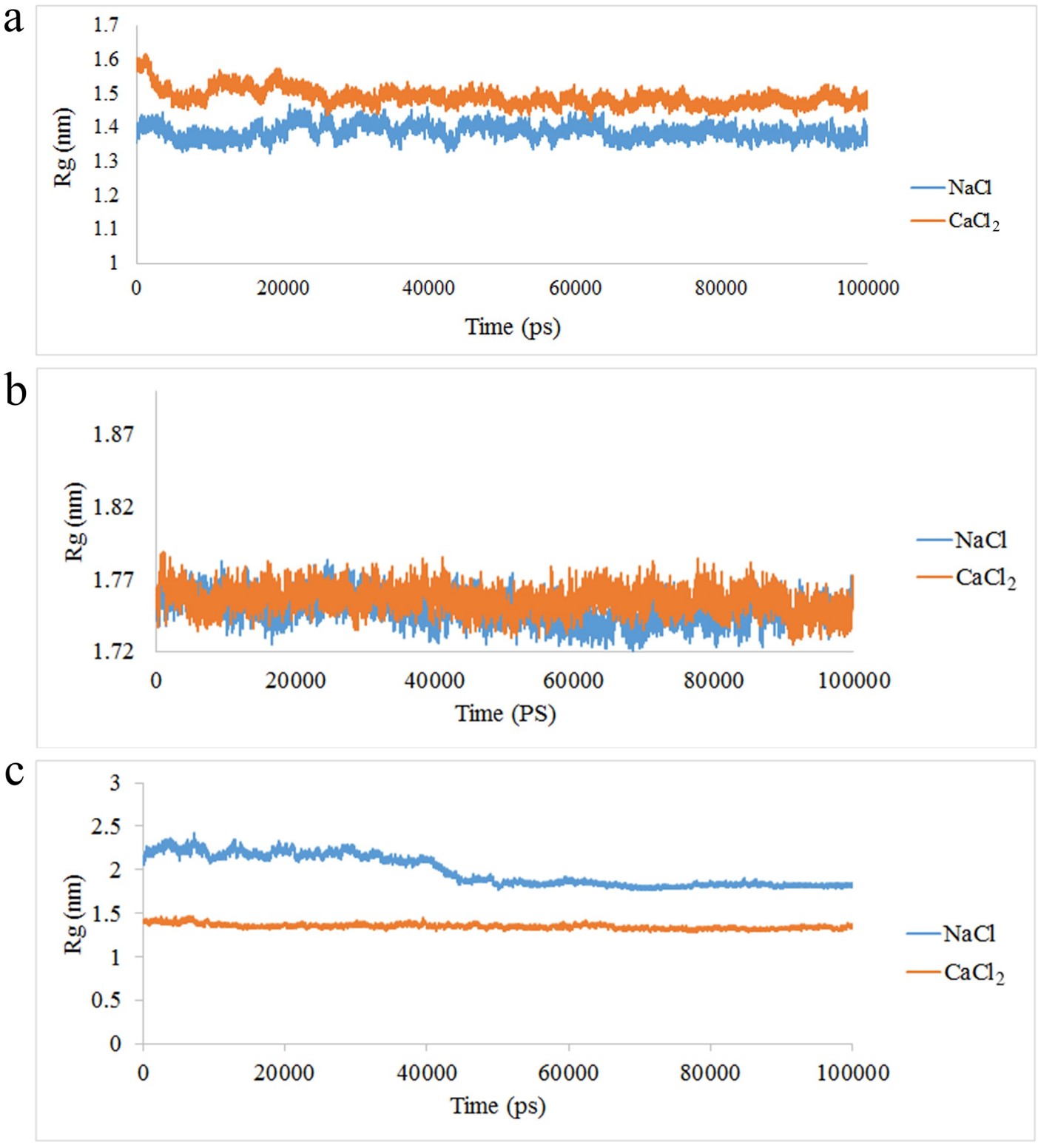

**Fig 2.** The Rg value of (a) A9, (b) A8 and (c) A8/A9 protein in presence of $Na^+$ and $Ca^{2+}$.

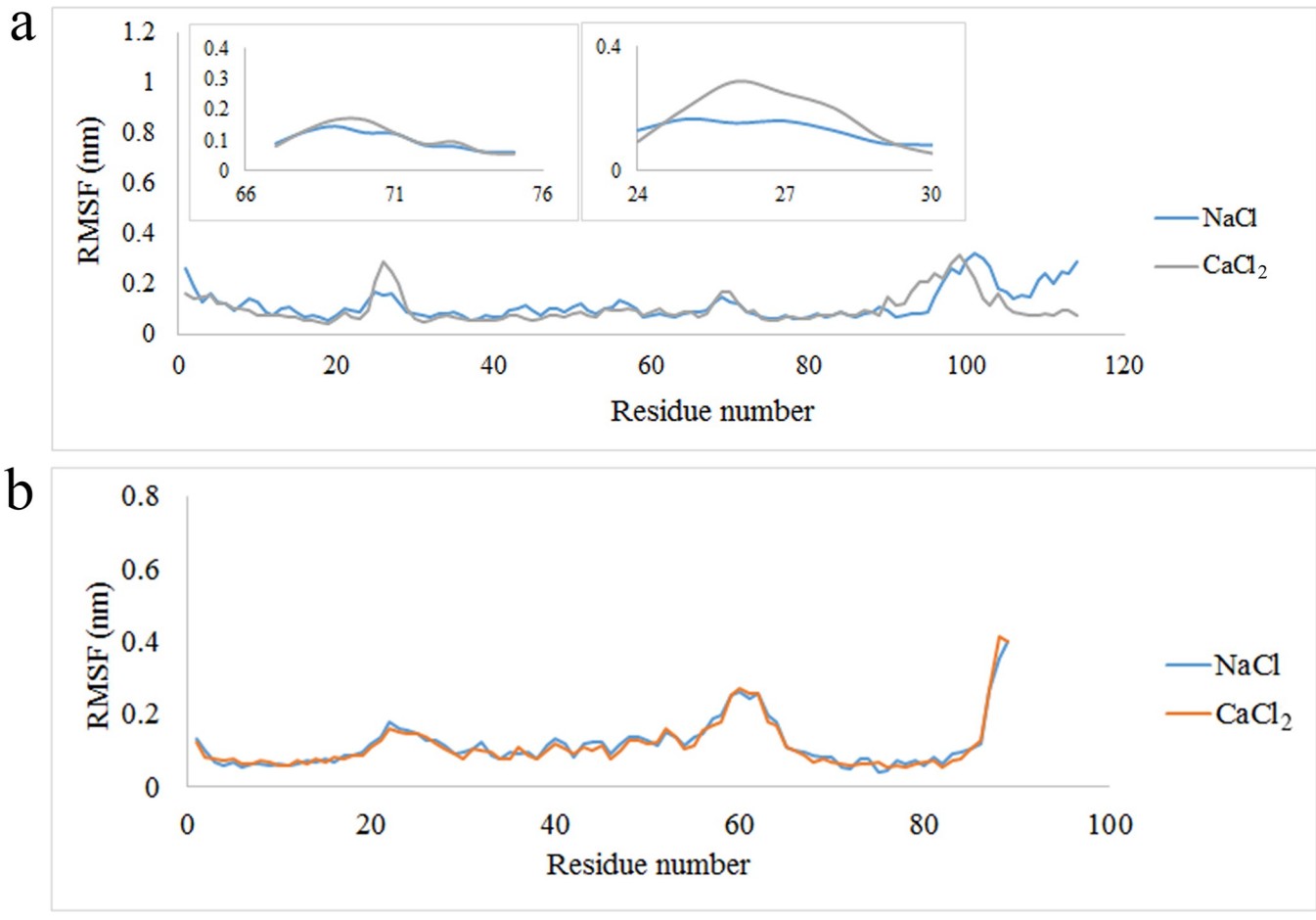

**Fig 3.** (a) The RMS fluctuation value of A9 protein in presence of $Na^+$ and $Ca^{2+}$. Inset: the A9 protein has been reached the higher flexibility in 25–30 and 68–71 amino acid regions in presence of $Ca^{2+}$ than $Na^+$ (b) The RMS fluctuation value of A8 protein in presence of $Na^+$ and $Ca^2$.

that presence of $Ca^{2+}$ in system and interactions with these parts lead to structural changes of the A9 protein.

The RMSF value of the A8 protein in presence of $Na^+$ and $Ca^{2+}$ was almost similar (Fig 3B).

The RMSF value of the complex in presence of both ions was shown in Fig 4. The figure showed the comparison of each subunit of the complex in presence of $Na^+$ and $Ca^{2+}$. All the amino acids of the A8 subunit were less flexible in presence of $Ca^{2+}$ than $Na^+$ (Fig 4A). Furthermore, the main amino acid in A9 subunit were less flexible in presence of $Ca^{2+}$ than $Na^+$ (Fig 4B). These results verify that in presence of $Ca^{2+}$, the subunits undergo a wide interaction.

The SASA method could be applied to evaluate and compare various molecules and conformations and determining the surface which concealed because of oligomerization [40]. Moreover, it indicates the accessible surface of the protein solvent or a part of a protein which exposes to solvent [41]. The SASA profile of A9 protein was illustrated in Fig 5A. Based on the figure, the amino acids of A9 protein (80–90 residues) were exposed in presence of $Ca^{2+}$. In other words, the accessible surface of protein interaction region in presence of $Ca^{2+}$ was higher than sodium ion. This parameter in A8 protein in presence of both ions was almost similar (Fig 5B). The accessible surface of the complex in presence of $Ca^{2+}$ was significantly lower than the time which the complex was in $Na^+$ environment (Fig 5C). Altogether, the RMSD,

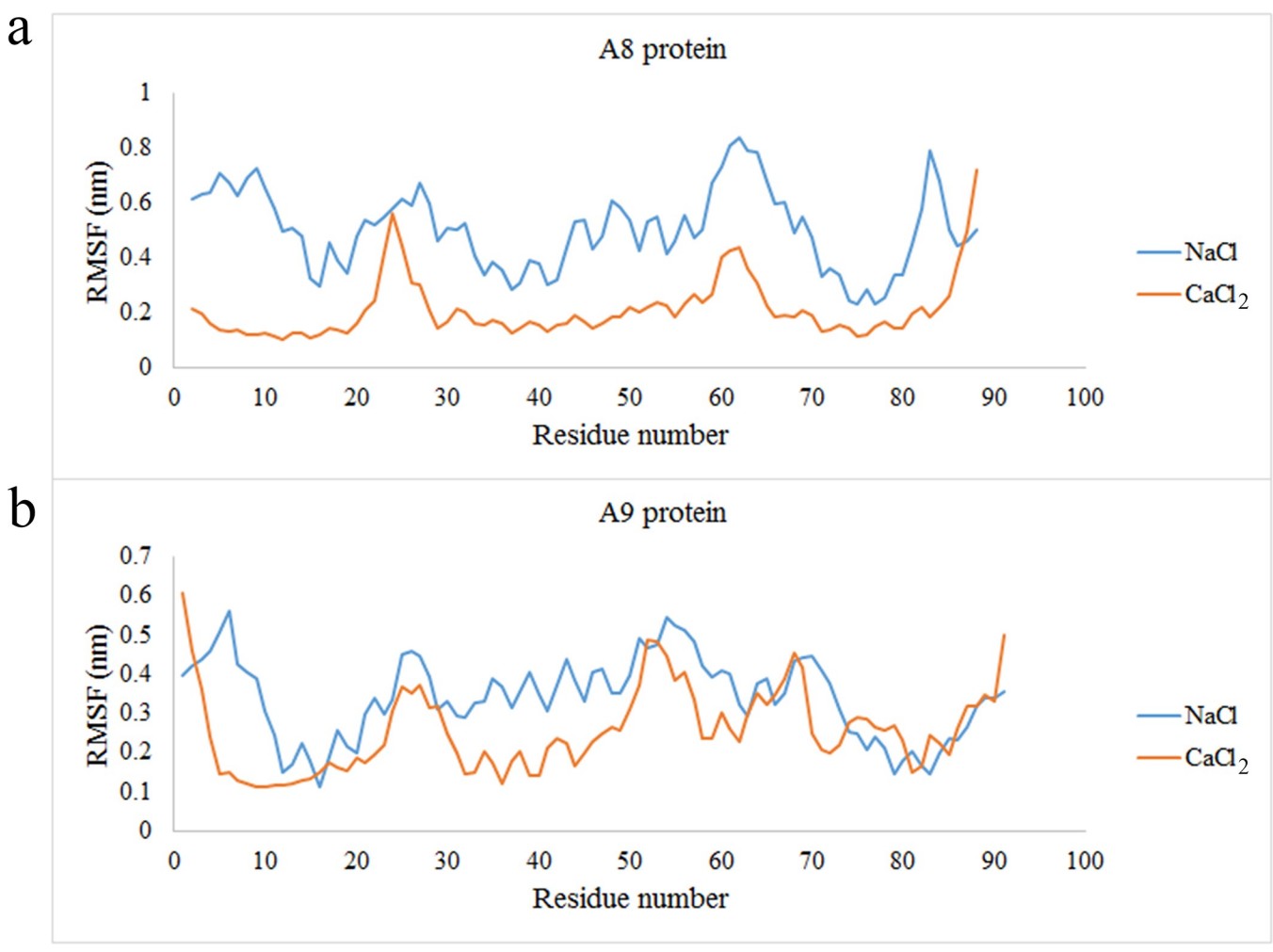

**Fig 4.** (a,b)The RMSF of A8/A9 complex in presence of Na$^+$ and Ca$^{2+}$.

Rg, RMSF and SASA results confirm an extensive structural changes of the complex in presence of Ca$^{2+}$ toward Na$^+$.

The amino acids of A9 protein which were exposed in presence of calcium ion were shown in Fig 6.

### RIN analysis

RIN, is a protein representation network which can provide the inter-residue and has a major role in proteins structural stability and functions. Hence, the RIN method could be used to evaluate the modifications in protein interaction network [42, 43]. In this study, the amino acids and their interactions were defined as nodes and edges. To provide the internal interaction network of the protein amino acids, the Ring program was used [23]. The resulted changes in internal interaction network of the complex in presence of Ca$^{2+}$ and Na$^+$ were shown using the program. In internal interaction network of the complex, the interactions of GLU4 and HIS91 were available in presence of Na$^+$ while the broad interactions between the amino acids of the A9 and A8 were observable in presence of Ca$^{2+}$ (S1 Fig). The analysis of the Calprotectin internal interaction network demonstrates an increased interaction between the

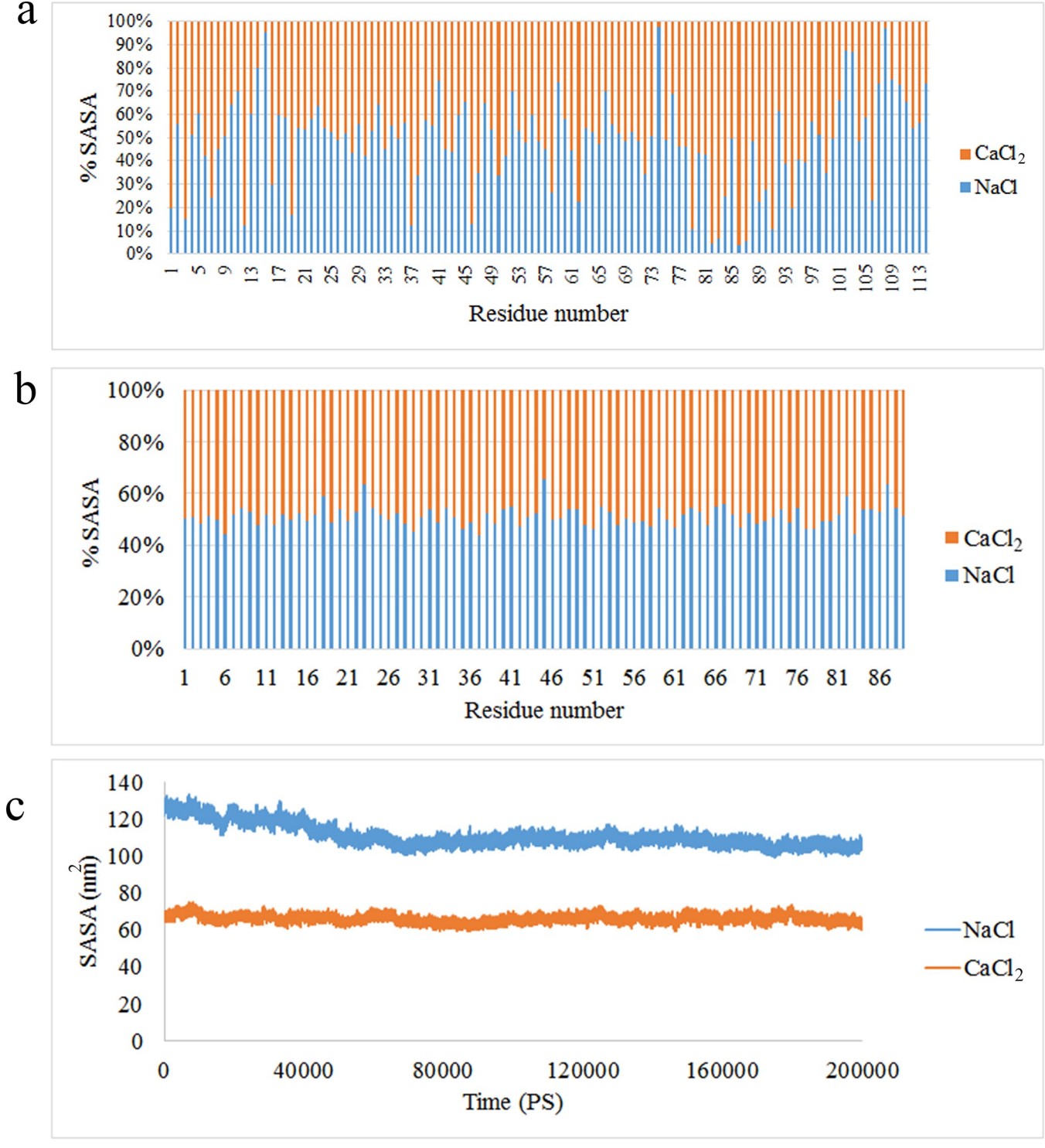

**Fig 5.** SASA value of **(a)** A9 protein (b) A8 and (c) A8/A9 complex in presence of Na$^+$ and Ca$^{2+}$.

A9 and A8 subunits in presence of Ca$^{2+}$. It seems that the availability of the calcium leads to A9 structural modifications and the accessibility of the required amino acids for stable A9-A8 complex production.

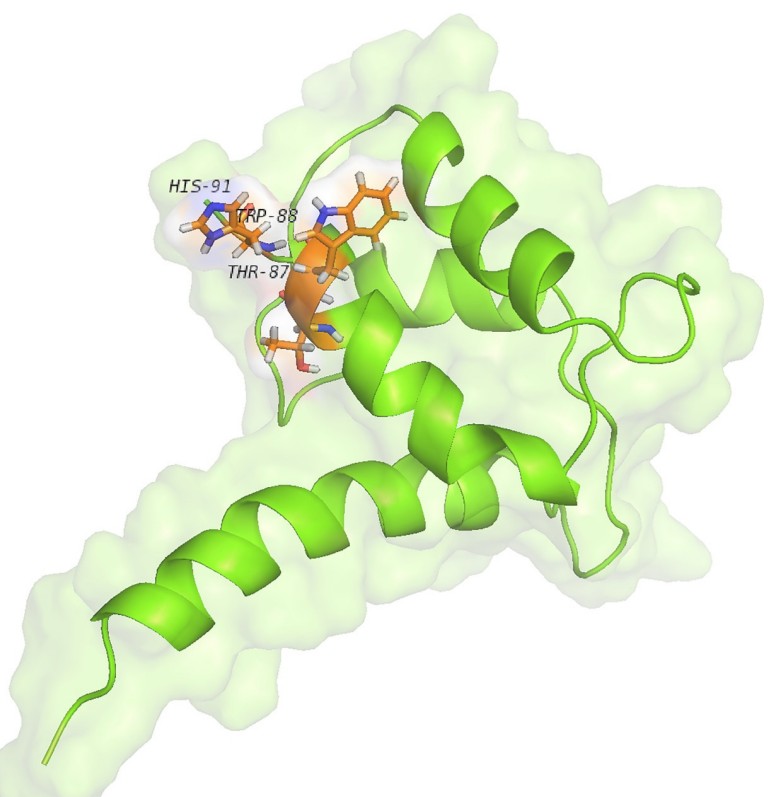

**Fig 6.** The structural modifications of A9 protein in presence of Ca$^{2+}$. The figure depicts the amino acids which involved in interaction of two subunits: His91, Trp88, Thr87.

### Influence of ions on Calprotectin association

The US is a method which has the ability to calculate the free energy extensively and has long be applied to evaluate the ligand-receptor separation procedures and binding free energy determination [44, 45]. To recognize the interaction between two subunits in various ionic conditions, the PMF value was calculated. The PMF results of each ionic conditions were illustrated in Fig 7. As demonstrated, the binding energy of PMF indicates that the Calprotectin complex is more stable in presence of Ca2$^{+}$ than Na$^{+}$. Moreover, the binding energy of two subunits (A9 and A8) in presence of Ca$^{2+}$ and Na$^{+}$ was 23.4 and 13.4 kcal/mol respectively. The results verify that the binding energy of Calprotectin subunits in presence of Ca$^{2+}$ is approximately 2 fold stronger than the time which the complex is in sodium solution. Hence, the binding energy of PMF is in consistent with MMPBSA results with similar report. Additionally, in presence of Ca$^{2+}$, the Calprotectin subunits undergo structural changes which result in stronger reactions of the complex in compare with Na$^{+}$. Furthermore, previous studies indicated that in presence of Ca$^{2+}$, the hydrophobic amino acids of Calprotectin subunits became more accessible [9] which in current study, the structural modifications of Calprotectin subunits resulted in new amino acids accessibility. Thus, new interactions between the subunits were formed.

### Binding energy of Calprotectin complex

In previous studies, the MM/PBSA method was considered to evaluate the ligand and receptor activity during the MD simulation [46, 47]. To evaluate the role of Ca$^{2+}$ and Na$^{+}$ in binding

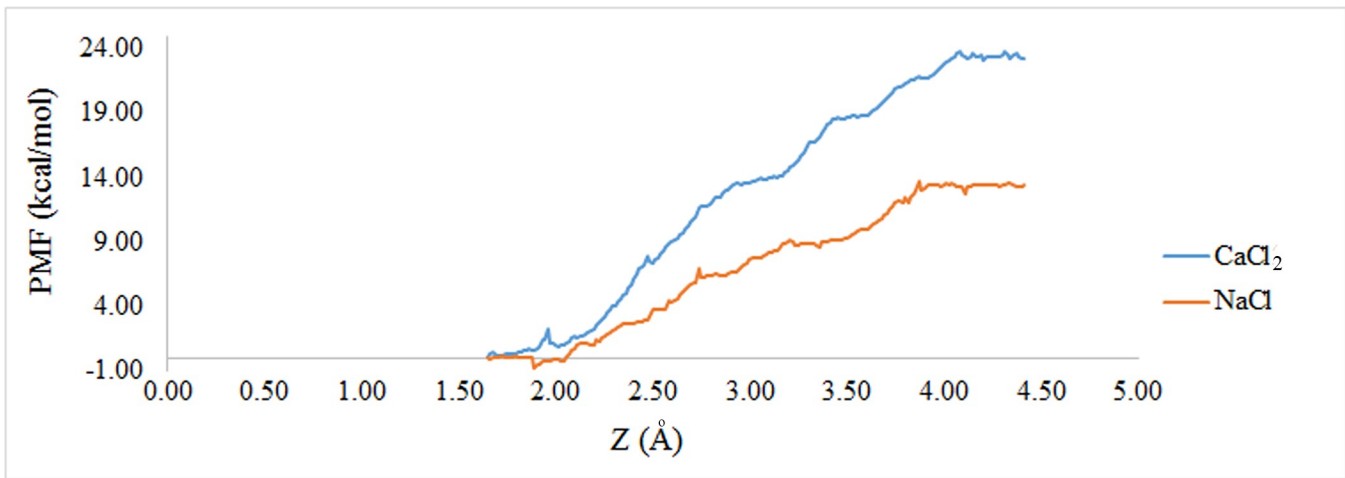

**Fig 7. The binding free energy profile of A8 protein dissociated from A9 in calprotectin complex in presence of Na⁺ and Ca²⁺.**

two subunits to each other, their binding energy was calculated using MM/PBSA method. The results demonstrated that in presence of $Ca^{2+}$, the binding energy was significantly higher than the time which the complex was in presence of $Na^+$. The binding energy of the Calprotectin subunits in presence of $Ca^{2+}$ and $Na^+$ were -287.292 kJ/mol and -194.388 kJ/mol respectively. In one study Korndörfer IP et al confirmed that in presence of $Ca^{2+}$, the LEU9, ILE12, ILE13, PHE68, GLN69, and LEU72 amino acids of A8 and the THR87, TRP88, and HIS91 amino acids of A9 protein play an important role in binding the two subunits to each other [9]. These findings indicate the role of hydrophobic amino acids in binding two subunits to each other. Furthermore, in presence of $Ca^{2+}$, the hydrophobic amino acids were more accessible and a network of hydrophobic interactions were created between the two subunits which led to a higher stability of the complex. Fig 8, illustrates that in presence of $Na^+$, only the side chain of A8 protein (LEU9) locates in interaction region. However, in presence of $Ca^{2+}$, the A9 moves toward A8, therefore the interaction region of heterodimer including THR87, TRP88,

a                              b

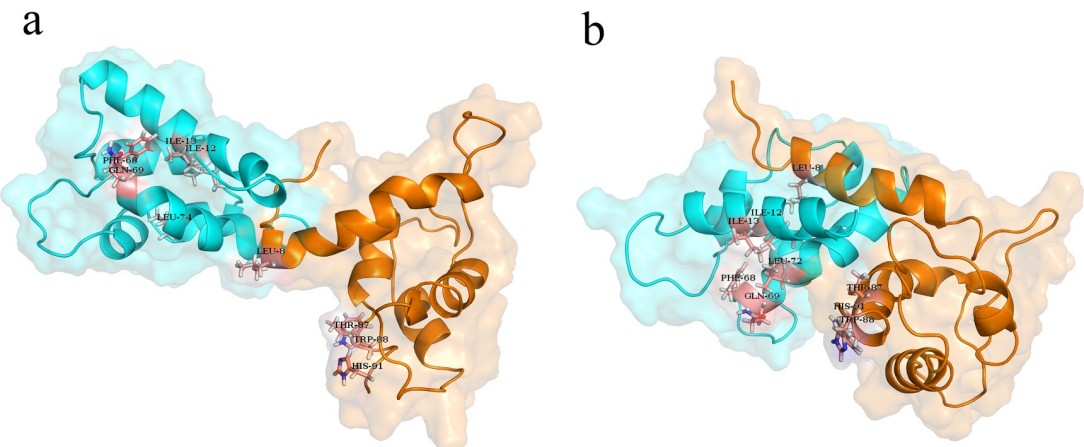

**Fig 8. (a)** The structural modifications of the calprotectin complex in presence of Na⁺ and **(b)** Ca²⁺.

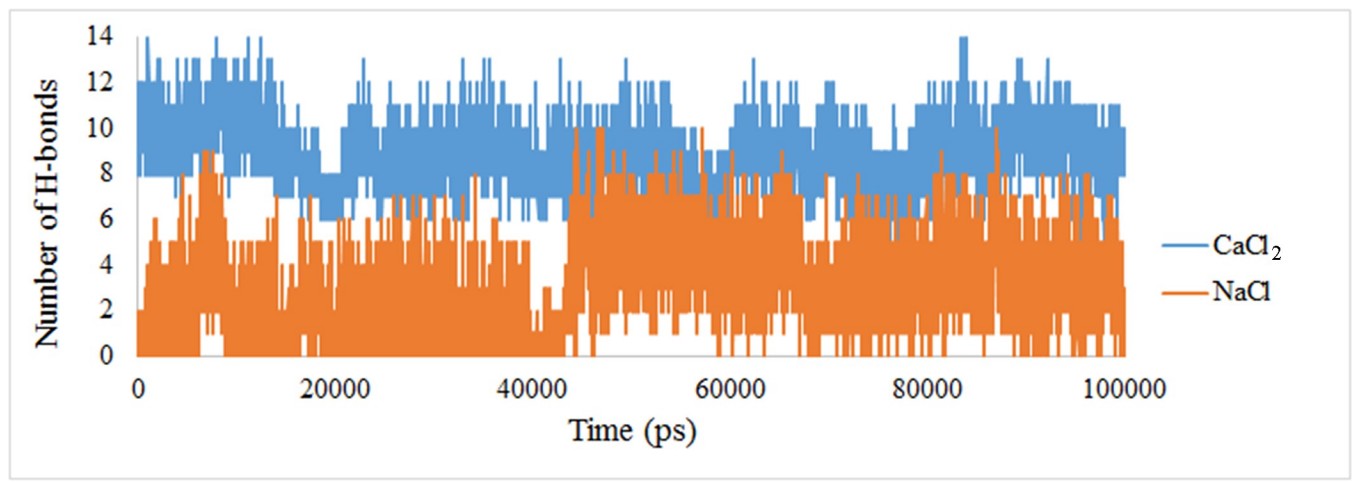

**Fig 9. The number of H-bonds between A8 and A9 subunits in presence of Na⁺ and Ca²⁺.**

HIS91 (from A9 protein) and the GLN69 and LEU72 (from A8 protein) become closer to each other.

## Influence of ionic conditions on Hydrogen bond interactions of the Calprotectin

The protein intermolecular H-bond formation plays a fundamental role in systems stability. The H-bond network is the exclusive solvent with the capability to monitor the dynamics, stability, structure and also the biomolecular functions [48]. The average number of H-bonds during the simulation when the heterodimer was present in presence of sodium and calcium ions were 3 and 11 respectively. The increase in the number of H-Bonds in presence of calcium indicates a stronger linkage and a higher stability of heterodimer in this manner (Fig 9).

## Calcium entrance in to the A9 structure

The distance of calcium ion from A9 subunit was shown in Fig 10. As depicted in the figure, after 30 ns, the calcium ion was located at 0.6 nm distance of loop I (Fig 10A) and this interval was stable during the simulation. Another calcium ion was located at the interval of 0.6 nm from loop II after 15 ns and the distance was constant during the simulation. The 3D image of the calcium ions position in loop I and II was illustrated in Fig 10B. The calcium entrance in loop II in a short time may be due to the presence of negative charged amino acids which are abundant in loop II. (2 Asp in loop II in comparison with 1 Asp in loop I) [49].

Several studies proved that the calcium can induce conformational changes in protein subunits and facilitate the dimerization process [8, 49, 50]. The 3D structure of the subunits and the complex in presence of both ions was shown in Fig 11. In our study, it was demonstrated that the calcium ion can enter the structure (Fig 11A) which was confirmed by previous studies [4] but sodium possesses no binding site for the A9 protein (Fig 11B). The Fig 11C, illustrates the calcium availability in loop I and II of A8 subunit. Furthermore, the A8/A9 complex was shown in presence of Na⁺ and Ca²⁺ (Fig 11D and 11E). It should be noted that the calcium ion can induce the structural changes of the subunits which lead to a better biological activity of the complex. Moreover, less structural changes in NaCl in comparison to CaCl2 for A9 protein and the A8/A9 complex was observed.

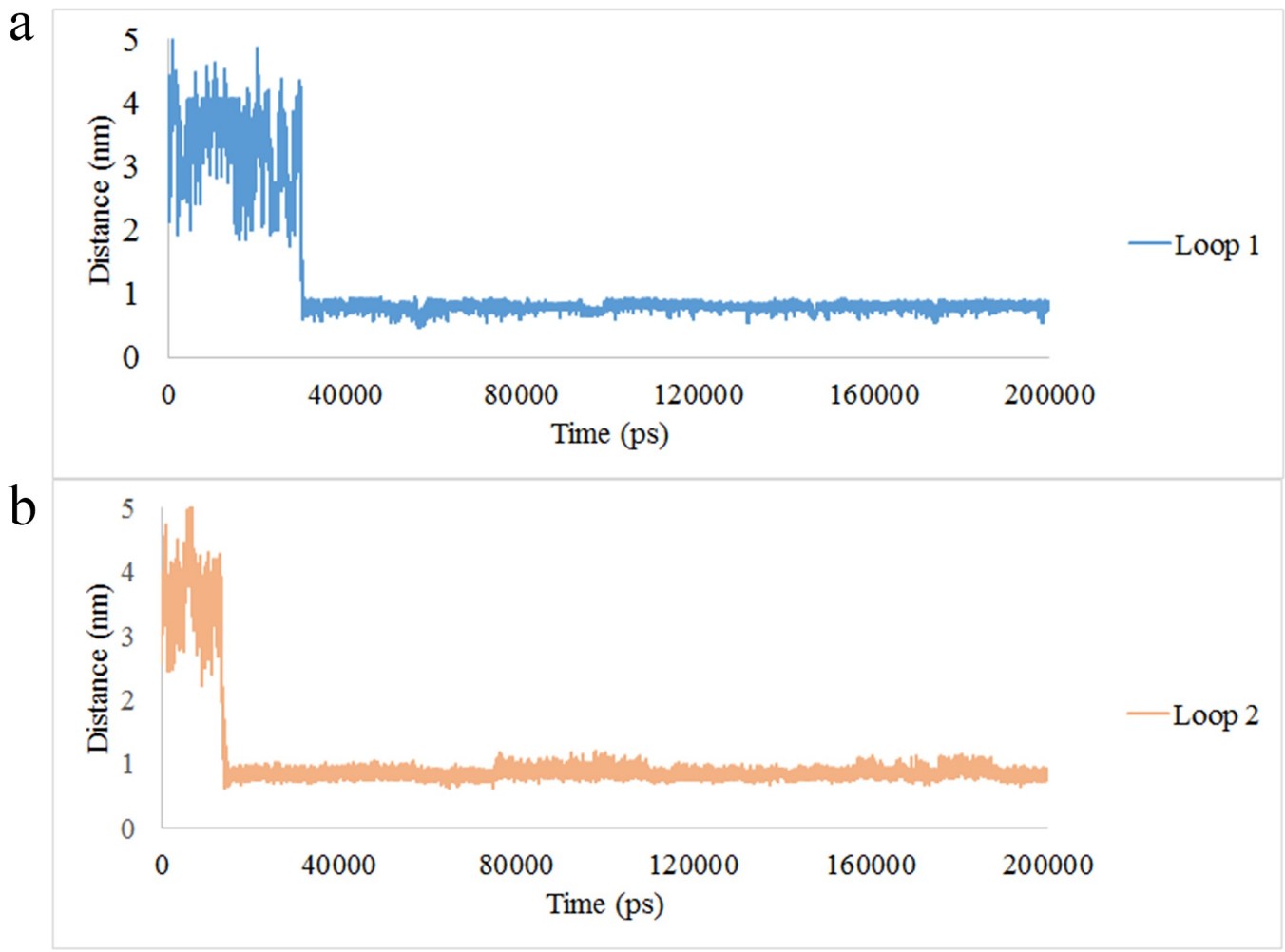

**Fig 10.** The distance of calcium ion from (a) loop I and (b) loop II of the A9 protein.

## Conclusion

The Calprotectin has a key role in various kinds of disease including cancers, autoimmune disorders and the involved processes such as inflammation. Hence, a better recognition of the protein and the mechanism of action is needed. In this study, we aimed to evaluate and characterize the protein structure in different ionic conditions. Thus, the MD simulation method was used to study the structural changes of the protein complex in presence of $Na^+$ and $Ca^{2+}$. The results represented a stronger binding and higher stability of the Calprotectin subunits in presence of $Ca^{2+}$ which was the consequence of structural changes of the subunits. Moreover, such parameters including: RMSD, RMSF, SASA, RG and H-bonds confirmed the structural changes, while the PMF and MM-PBSA verified the stronger binding of the protein subunits. The results of this study confirm that presence of calcium leads to structural changes of the Calprotectin and eventually the accessibility of hydrophobic amino acids for stronger interactions and higher stability of the complex which declare the importance of ionic conditions in intracellular inflammation procedures. Hence, these findings could be useful in protein structure recognition which could be applied in drug design, diagnosis and targeted treatments of the disease in the future.

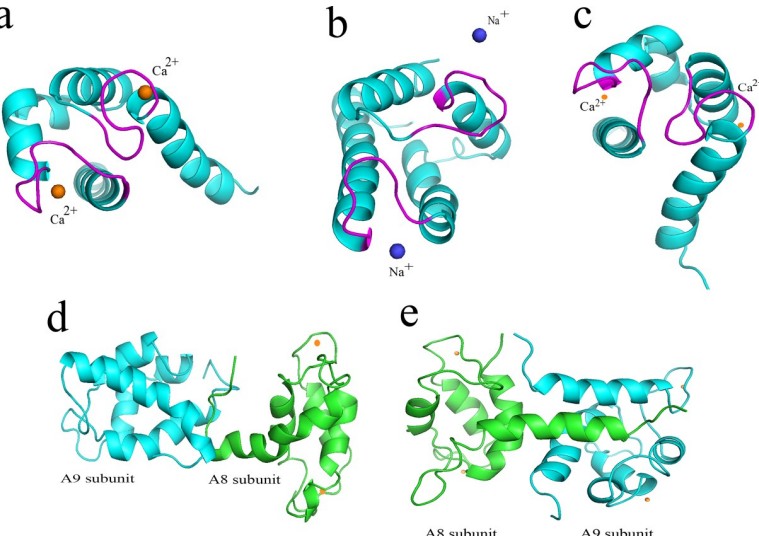

**Fig 11.** The 3D structure of the (a) A9 in presence of $Ca^{2+}$, (b) A9 in presence of $Na^+$ (c) A8 in presence of $Ca^{2+}$ (d) A8/A9 complex in presence of $Na^+$ (e) A8/A9 complex in presence of $Ca^{2+}$.

## Supporting information

**S1 Fig.** (a) The residue interaction network (RIN) between the calprotectin complex in presence of $Na^+$ and (b) $Ca^{2+}$. As illustrated, an increased interaction between the A9 and A8 subunits in presence of $Ca^{2+}$ is observable. Green (A9), red (A8).
(TIF)

**S1 Table. Summary of the MD runs.**
(DOCX)

## Acknowledgments

The authors appreciate the Research Council of Qazvin University of Medical Sciences.

## Author Contributions

**Formal analysis:** Mohammad Ghorbani.

**Investigation:** Mohammad Ghorbani.

**Methodology:** Alireza Farasat.

**Project administration:** Alireza Farasat.

**Writing – original draft:** Nematollah Gheibi.

**Writing – review & editing:** Hanifeh Shariatifar.

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
