## [Decision Letter · Decision Letter 0]

19 Aug 2019

PONE-D-19-21711

In silico assessment of human Calprotectin subunits (S100A8/A9) in presence of sodium and calcium ions using Molecular Dynamics simulation approach

PLOS ONE

Dear Dr. Farasat,

Thank you for submitting your manuscript to PLOS ONE. After careful consideration, we feel that it has merit but does not fully meet PLOS ONE’s publication criteria as it currently stands. Therefore, we invite you to submit a revised version of the manuscript that addresses the points raised during the review process.

ACADEMIC EDITOR: Please try to improve your manuscript according to the very serious criticism of the reviewers. 

We would appreciate receiving your revised manuscript by Oct 03 2019 11:59PM. To enhance the reproducibility of your results, we recommend that if applicable you deposit your laboratory protocols in protocols.io, where a protocol can be assigned its own identifier (DOI) such that it can be cited independently in the future. For instructions see: http://journals.plos.org/plosone/s/submission-guidelines#loc-laboratory-protocols

We look forward to receiving your revised manuscript.

Kind regards,

Eugene A. Permyakov, Ph.D., Dr.Sci.

Academic Editor

PLOS ONE

**Journal Requirements:**

**Comments to the Author**

1. Is the manuscript technically sound, and do the data support the conclusions?

Reviewer #1: Partly

Reviewer #2: Partly

2. Has the statistical analysis been performed appropriately and rigorously? 

Reviewer #1: I Don't Know

Reviewer #2: No

3. Have the authors made all data underlying the findings in their manuscript fully available?

Reviewer #1: Yes

Reviewer #2: No

4. Is the manuscript presented in an intelligible fashion and written in standard English?

Reviewer #1: Yes

Reviewer #2: No

5. Review Comments to the Author

Reviewer #1: The protein object of research and the scientific task posed in the manuscript are interesting. It is also important that several methods were used to solve it.

My main comments are related to the choice of PDB files used in the calculations.

1) Why was the PDB file 5I8N for protein A9 taken?

The authors write that it is apo-protein. In the article that accompanies this structure there is not a word about the apo-form. It is possible that the absence of metal atoms in this structure of A9 is due to the fact that the conformation was determined using NMR method. In section 3.1, this protein was analyzed in the presence of calcium and sodium. How metal atoms were placed in the protein structure? It is necessary to write about this in the Methods section.

2) Why was the PDB file 5HLO for protein A8 taken?

This structure contains atoms of calcium, zinc and chlorine. The work analyzed the structure with sodium. Why the structure with calcium was not analyzed, as was done for A9? What was done with the atoms of zinc and chlorine?

3) A PDB file 1XK4 was taken for structural analysis of the complex A8/A9. This is the right choice. Perhaps it was worth using only this single file for all the work, extracting monomers from the dimer when necessary.

Minor comments:

1) Check phrases:

“In the next step, (The system was neutralized by 150 mM NaCl and CaCl2) [4, 18]“

“The primary structure of the dimer was obtained by the A9 protein superimposition on the A9 subunit of the complex (ID: 1XK4)”.

2) “3.1. Analysis of structural and conformational properties of all systems”

What is the difference between words “structural and conformational”?

Which means “all systems”?

3) Check phrase:

“(the average of RMSD value for A9 in presence of Ca 2+ and Na + were calculated at last 20 ns of simulations)”.

4) No references to Protein Data Bank or to publications mentioned in the PDB files of A8, A9 and their complex.

Reviewer #2: The authors present a paper studying the effects of sodium and calcium ions on the structure of human Calprotein.

They employed molecular dynamics simulations to investigate structural differences in presence of biological concentration of Ca2Cl and NaCl. The authors aim to address an interesting question and molecular dynamics is an appropriate method to study structural protein changes due to presence of different ions. However, I can not recommend this manuscript for publication in present state. First of all, I can not judge the quality and accuracy of the simulations because important details about system preparation, equilibration, and MD protocol were omitted (see point 1). Additionally, the manuscript lacks rigorous analysis and comparison to what is known experimentally (see points 2 and 3). Finally, this is a paper looking at Calprotein in presence of sodium and calcium ions, but there is no analysis of ion behavior (point 4). Please see the comments below on how to improve this paper.

1. System preparation and equilibration. The authors do mention equilibration in the Methods section but they do not provide any details on how it was done. When using molecular dynamics it is common practice to restrain the protein structure while equilibrating water and ions for a shot time. It is not clear if this was done for the simulations shown here, and skipping this equilibration step can result in large deviations from the crystal structure due to initially unfavorable water/ion positions. Additionally, structure 1XK4 used in the dimer simulation contains several crystal water molecules and calcium ions. Did the authors keep crystal water and ions for all structures? What did they do with the crystal calcium ions in sodium simulations? Finally, several details from MD protocol are omitted: time step, bond restraints, cutoffs and long range electrostatics.

2. A rigorous comparison of all studied structures with both two ions is needed. The authors did significant structural analysis for the A9 monomer (RMSF, SASA and so on), but not for the A8 monomer, or the A8/A9 dimer. This analysis needs to be performed on all three systems and with both ions (6 total systems). Based on the presented date, the authors can only claim that structural changes were observed in presence of calcium for the A9 monomer. The visual comparison of dimer structures with sodium vs calcium in Figure 8 is appreciated and it should be done for the monomer structures as well. Finally, all equilibrium simulations should be repeated at least ones to ensure that the observed structural changes are consistent.

3. Comparison to experiments. It is important to compare simulations to what is known experimentally. The authors should discuss if the used crystal structure were prepared in presence of sodium, calcium or a different ion. If a system was prepared in NaCl I would expect less structural changes in NaCl in comparison to Ca2Cl. Was this observed for the simulated systems? In addition, the calculated magnitude of dimer stabilization should be compared to what is observed experimentally.

4. Analysis of ion behavior. One major advantage of studying ion effects with MD is the ability to analyze ion behavior in detail. Unfortunately, this was not analyzed in this manuscript. The authors should compare ion behavior and interactions with the protein for calcium and sodium systems.

Other notes:

The English needs to be improved. In present form the paper contains too many grammatical errors and unclear sentences for reviewer corrections.

The authors should add a figure showing the different domains of Calprotein and where the calcium ions are binding, as well as a table of all simulated systems.

References should be correctly formatted. Right now several references lack journal name, and/or volume, or pages.

6. PLOS authors have the option to publish the peer review history of their article (what does this mean?). If published, this will include your full peer review and any attached files.

Reviewer #1: No

Reviewer #2: No

---

## [Author Response · Author response to Decision Letter 0]

3 Oct 2019

Response to the Reviewers:

We truly appreciate the reviewer’s valuable and constructive comments. We have adopted all the suggestions in our revised manuscript.

Reviewer: 1

The protein object of research and the scientific task posed in the manuscript are interesting. It is also important that several methods were used to solve it. My main comments are related to the choice of PDB files used in the calculations.

1) Why was the PDB file 5I8N for protein A9 taken?

The authors write that it is apo-protein. In the article that accompanies this structure there is not a word about the apo-form. It is possible that the absence of metal atoms in this structure of A9 is due to the fact that the conformation was determined using NMR method. In section 3.1, this protein was analyzed in the presence of calcium and sodium. How metal atoms were placed in the protein structure? It is necessary to write about this in the Methods section.

Reply:

In RCSB Protein Data Bank (PDB) two files exist for S100A9 subunit: 1. PDB ID: 1IRJ, which is in complex with CHAPS molecule. The interaction of S100A9 with CHAPS molecule results in some structural changes in A9 protein which makes it an inappropriate candidate for evaluating the structural changes. 2. PDB ID: 5I8N, in this file the S100A9 protein is alone which makes it a suitable candidate for evaluating the structural changes of the protein which caused by calcium ions and also to improve the dimerization process. Of course the conformation was determined by NMR method, but to observe the effects of calcium ions on the protein structural changes, the system was neutralized by Na+ and Cl-. Then, the concentration of 150 mM NaCl and CaCl2 were added to each system. The results revealed that the sodium ions could not enter the loops but the calcium ions entered successfully and made the structural changes of the protein (Fig 12a, b). 

2) Why was the PDB file 5HLO for protein A8 taken?

This structure contains atoms of calcium, zinc and chlorine. The work analyzed the structure with sodium. Why the structure with calcium was not analyzed, as was done for A9? What was done with the atoms of zinc and chlorine?

Reply:

As S100A9, there were no PDB files without structural ions for S100A8. So, we had to select a PDB file which contained all the ions which were needed for inducing the structural changes and evaluate the S100A8/A9 heterodimerization. Moreover, this PDB file contains all the ions (zinc, chloride and etc.) so we found out the 5HLO a suitable PDB file for this evaluation. Because in this condition the presence of these ions lead to a better structural changes and heterodimerization. As mentioned in the manuscript, the simulation process was done with all the ions (zinc, chloride, etc.) in NaCl as the control and also in revision step, to verify that the structural changes were occurred by the ions, the simulation was repeated in CaCl2 as a model. The results showed that the S100A8 in both systems possesses similar structures which confirm the above findings.

3) A PDB file 1XK4 was taken for structural analysis of the complex A8/A9. This is the right choice. Perhaps it was worth using only this single file for all the work, extracting monomers from the dimer when necessary.

Reply:

We didn’t use 1XK4 for all the work, because as mentioned before, firstly we aimed to induce structural changes in A8 and A9 and secondly we evaluated their dimerization. Thus, the orientation of this complex (A8/A9, ID: 1XK4) was used to locate the above subunits (ID: 5I8N, ID: 5HLO) appropriately besides each other.

Minor comments:

1) Check phrases:

“In the next step, (The system was neutralized by 150 mM NaCl and CaCl2) [4, 18]“

“The primary structure of the dimer was obtained by the A9 protein superimposition on the A9 subunit of the complex (ID: 1XK4)”.

Reply:

Firstly the system was neutralized by Na+ and Cl- . Then, the concentration of 150 mM NaCl and CaCl2 were added to the systems.

The orientation of this complex (A8/A9, ID: 1XK4) was used to locate the above subunits (ID: 5I8N, ID: 5HLO) besides each other.

2) “3.1. Analysis of structural and conformational properties of all systems” What is the difference between words “structural and conformational”? Which means “all systems”?

Reply:

In section 3.1, We made a mistake about structural and conformational words. Thus it was edited. The structural word is the correct one.

In this part all systems mean: 

The A9 protein in presence of sodium and calcium ions

The A8 protein in presence of sodium and calcium ions

The A8/A9 complex protein in presence of sodium and calcium ions

3) Check phrase:

“(the average of RMSD value for A9 in presence of Ca 2+ and Na + were calculated at last 20 ns of simulations)”.

Reply:

This phrase was omitted.

4) No references to Protein Data Bank or to publications mentioned in the PDB files of A8, A9 and their complex.

Reply:

The references were added to the revised manuscript.

Reviewer 2: 

1.System preparation and equilibration. The authors do mention equilibration in the Methods section but they do not provide any details on how it was done. When using molecular dynamics, it is common practice to restrain the protein structure while equilibrating water and ions for a short time. It is not clear if this was done for the simulations shown here, and skipping this equilibration step can result in large deviations from the crystal structure due to initially unfavorable water/ion positions. Additionally, structure 1XK4 used in the dimer simulation contains several crystal water molecules and calcium ions. Did the authors keep crystal water and ions for all structures? What did they do with the crystal calcium ions in sodium simulations? Finally, several details from MD protocol are omitted: time step, bond restraints, cutoffs and long range electrostatics.

Reply:

In current study, the MD simulation process was performed using GROMACS program version 5.1 and the CHARMM 36 force field was used for all simulations. The A9 (ID: 5I8N) and A8 (ID: 5HLO) proteins were obtained by RCSB Protein Data Bank (PDB). In this study, the above systems were solvated by transferable intermolecular potential with 3 points (TIP3P) water model in a cubic box with a distance of 10 Å from the furthest atom of the protein. After solvation, Na+ and Cl− ions were inserted to neutralize the system. Then, the concentration of 150 mM NaCl and CaCl2 were added to the systems and the energy minimization was carried out using the steepest descent method. Each system was equilibrated by 1 ns MD simulation in the canonical (NVT) ensemble and 1 ns MD simulation in the isothermal–isobaric (NPT) ensemble using position restraints on the heavy atoms of the protein to allow for the equilibration of the solvent. The Nose–Hoover thermostat constant was used for fixing the temperature of the system at 310 K. To maintain the pressure of the system at fixed 1 bar pressure, the Parrinello–Rahman pressure coupling method was used. The electrostatic interactions were calculated using the Particle Mesh Ewald (PME) method with 1.0 nm short-range electrostatic and van der Waals cutoffs. To evaluate the effects of calcium ion on dimerization process, the A9 with and without calcium was used. Thus, the orientation of the complex (A8/A9, ID: 1XK4) was applied to locate the above subunits (ID: 5I8N, ID: 5HLO) appropriately besides each other. Consequently, the process of 100 and 200 ns MD simulation was carried out for dimer (A8/A9) and monomers (each subunits) with time steps of 2 fs on equilibrated systems respectively. 

The “posre.itp” file was generated by pdb2gmx; it defines a force constant used to keep atoms in place during equilibration. So, it was done as a single step during the MD simulation process. It should be noted that we didn’t use 1XK4 for all the work, because as mentioned before, firstly we aimed to induce structural changes in A8 and A9 and secondly we evaluated their dimerization. Thus, the orientation of this complex (A8/A9, ID: 1XK4) was used to locate the above subunits (ID: 5I8N, ID: 5HLO) besides each other. The simulation of the systems in presence of sodium and calcium was performed with the structural ions. 

2. A rigorous comparison of all studied structures with both two ions is needed. The authors did significant structural analysis for the A9 monomer (RMSF, SASA and so on), but not for the A8 monomer, or the A8/A9 dimer. This analysis needs to be performed on all three systems and with both ions (6 total systems). Based on the presented date, the authors can only claim that structural changes were observed in presence of calcium for the A9 monomer. The visual comparison of dimer structures with sodium vs calcium in Figure 8 is appreciated and it should be done for the monomer structures as well. Finally, all equilibrium simulations should be repeated at least ones to ensure that the observed structural changes are consistent.

Reply:

As depicted in Fig. 1a there was no difference between the RMSD value of A8 protein in presence of Na+ and Ca2+. It means that the RMSD value of the A8 protein in presence of Na+ and Ca2+ was nearly similar. The A8 protein in both systems was equilibrated after 10 ns. The Rg results of the aforementioned protein in presence of Na+ and Ca2+ was almost similar which revealed that the protein contraction in both conditions was similar (Fig. 2b). Moreover, the RMSF and SASA results of the A8 protein in both conditions were relatively similar (Fig. 3b and Fig. 5b). These findings show that the calcium ions induce the structural changes in the protein. In this study we aimed to use this protein for heterodimerization. 

The RMSD value of A8/A9 complex in presence of Na+ and Ca2+ was illustrated in Fig. 1c As shown in the figure, the complex was equilibrated in presence of Na+ after 10 ns and Ca2+ after 50 ns. The RMSD average of the complex in presence of Na+ and Ca2+ were 3 Å and 12 Å respectively. These findings confirm that the RMSD value of the complex in presence of Ca2+ was lower than the time which the complex was in sodium condition. This was due to the strong interaction between two subunits in presence of Ca2+ which led to the complex stabilization. The Fig. 2c demonstrates the Rg results in presence of Na+ and Ca2+. As illustrated, the complex was more condensed in presence of Ca2+ than Na+. The RMSF value of the complex in presence of Na+ and Ca2+ was shown in Fig. 4The figure showed the comparison of each subunit of the complex in presence of Na+ and Ca2+. All the amino acids of the A8 subunit were less flexible in presence of Ca2+ than Na+. Furthermore, the main parts of the amino acid regions in A9 subunit were less flexible in presence of Ca2+ than Na+. these results verify that in presence of Ca2+, the subunits possess a wide interaction. To evaluate the structural changes of the complex in presence of Na+ and Ca2+, the SASA was measured (Fig. 5c). As illustrated, the accessible surface of the complex in presence of Ca2+ was significantly lower than the time which the complex was in Na+ environment. Altogether, the RMSD, Rg, RMSF and SASA results confirm an extensive structural changes of the complex in presence of Ca2+ toward Na+. based on your valuable comment, the Fig. 12 Was added to the manuscript. The figure shows the 3D structure of the A8 and A9 subunits in presence of Na+ and Ca2+. Moreover, the simulation process was repeated for all six systems and the results changes were not remarkable in comparison with the first simulation. 

3. Comparison to experiments. It is important to compare simulations to what is known experimentally. The authors should discuss if the used crystal structure were prepared in presence of sodium, calcium or a different ion. If a system was prepared in NaCl I would expect less structural changes in NaCl in comparison to CaCl2. Was this observed for the simulated systems? In addition, the calculated magnitude of dimer stabilization should be compared to what is observed experimentally.

Reply:

For this PDB ID: 1xk4, the crystal structure was prepared in presence of chloride, calcium and citrate anion 

For this PDB ID: 5HLO, the crystal structure was prepared in presence of zinc, acetate, chloride, calcium and cacodylate ions.

For this PDB ID: 5I8N, the crystal structure was prepared in presence of NaCl and CaCl2

Regarding this fact; the sodium ion doesn’t have the binding site for calprotectin, so NaCl was used as a control for the system. Several studies proved that the calcium can induce conformational changes in protein subunits and facilitate the dimerization process [1-3]. In our study, it was demonstrated that sodium possesses no binding site for the A9 protein but the calcium ion can enter the structure. Furthermore, the calcium ion can induce the structural changes of the subunits which lead to a better biological activity of the complex. If a system was prepared in NaCl I would expect less structural changes in NaCl in comparison to CaCl2. Was this observed for the simulated systems? To answer this question, it should be noted that less structural changes in NaCl in comparison to CaCl2 for A9 protein and the A8/A9 complex was observed [1]. 

4. Analysis of ion behavior. One major advantage of studying ion effects with MD is the ability to analyze ion behavior in detail. Unfortunately, this was not analyzed in this manuscript. The authors should compare ion behavior and interactions with the protein for calcium and sodium systems.

Reply:

The behavior of the calcium ion with A9 subunit was shown in Fig. 11. As depicted in the figure, after 30 ns, the calcium ion was located at 0.6 nm distance of loop I and this interval was stable during the simulation. Another calcium ion was located at the interval of 0.6 nm from loop II after 15 ns and the distance was constant during the simulation. The 3D image of the calcium ions position in loop I and II was illustrated in Fig. 12a, c The calcium entrance in loop II in a short time may be due to the presence of negative charged amino acids which are abundant in loop II. (2 Asp in loop II in comparison with 1 Asp in loop I).

Other notes:

The English needs to be improved. In present form the paper contains too many grammatical errors and unclear sentences for reviewer corrections.

Reply: changes were done.

The authors should add a figure showing the different domains of Calprotein and where the calcium ions are binding, as well as a table of all simulated systems.

Reply: The aforementioned figure and the table of all simulated systems were added.

References should be correctly formatted. Right now several references lack journal name, and/or volume, or pages.

Reply: Reference changes were done

References

1. Gheibi N, Asghari H, Chegini K, Sahmani M, Moghadasi M. The role of calcium in the conformational changes of the recombinant S100A8/S100A91. Molecular Biology. 2016;50(1):118-23.

2. Leukert N, Vogl T, Strupat K, Reichelt R, Sorg C, Roth J. Calcium-dependent tetramer formation of S100A8 and S100A9 is essential for biological activity. Journal of molecular biology. 2006;359(4):961-72.

3. Streicher WW, Lopez MM, Makhatadze GI. Modulation of quaternary structure of S100 proteins by calcium ions. Biophysical chemistry. 2010;151(3):181-6.

---

## [Editor Report · Decision Letter 1]

7 Oct 2019

In silico assessment of human Calprotectin subunits (S100A8/A9) in presence of sodium and calcium ions using Molecular Dynamics simulation approach

PONE-D-19-21711R1

Dear Dr. Farasat,

We are pleased to inform you that your manuscript has been judged scientifically suitable for publication and will be formally accepted for publication once it complies with all outstanding technical requirements.

With kind regards,

Eugene A. Permyakov, Ph.D., Dr.Sci.

Academic Editor

PLOS ONE
---

## [Editor Report · Acceptance letter]

9 Oct 2019

PONE-D-19-21711R1 

In silico assessment of human Calprotectin subunits (S100A8/A9) in presence of sodium and calcium ions using Molecular Dynamics simulation approach 

Dear Dr. Farasat:

I am pleased to inform you that your manuscript has been deemed suitable for publication in PLOS ONE. Congratulations! Your manuscript is now with our production department. 

With kind regards,

on behalf of

Prof. Eugene A. Permyakov 

Academic Editor

PLOS ONE